# Treatment with *Glycyrrhiza glabra* Extract Induces Anxiolytic Effects Associated with Reduced Salt Preference and Changes in Barrier Protein Gene Expression

**DOI:** 10.3390/nu16040515

**Published:** 2024-02-13

**Authors:** Harald Murck, Peter Karailiev, Lucia Karailievova, Agnesa Puhova, Daniela Jezova

**Affiliations:** 1Department of Psychiatry and Psychotherapy, Philipps-University Marburg, 35039 Marburg, Germany; 2Institute of Experimental Endocrinology, Biomedical Research Center, Slovak Academy of Sciences, 845 05 Bratislava, Slovakia; karailiev.p@gmail.com (P.K.); lucia.karailievova@savba.sk (L.K.); agnesa.puhova@savba.sk (A.P.); daniela.jezova@savba.sk (D.J.)

**Keywords:** anxiety, depression, mineralocorticoid receptor, GFAP, claudin, zonula occludens protein-1, aldosterone

## Abstract

We have previously identified that low responsiveness to antidepressive therapy is associated with higher aldosterone/cortisol ratio, lower systolic blood pressure, and higher salt preference. *Glycyrrhiza glabra* (GG) contains glycyrrhizin, an inhibitor of 11β-hydroxysteroid-dehydrogenase type-2 and antagonist of toll-like receptor 4. The primary hypothesis of this study is that food enrichment with GG extract results in decreased anxiety behavior and reduced salt preference under stress and non-stress conditions. The secondary hypothesis is that the mentioned changes are associated with altered gene expression of barrier proteins in the prefrontal cortex. Male Sprague-Dawley rats were exposed to chronic mild stress for five weeks. Both stressed and unstressed rats were fed a diet with or without an extract of GG roots for the last two weeks. GG induced anxiolytic effects in animals independent of stress exposure, as measured in elevated plus maze test. Salt preference and intake were significantly reduced by GG under control, but not stress conditions. The gene expression of the barrier protein claudin-11 in the prefrontal cortex was increased in control rats exposed to GG, whereas stress-induced rise was prevented. Exposure to GG-enriched diet resulted in reduced ZO-1 expression irrespective of stress conditions. In conclusion, the observed effects of GG are in line with a reduction in the activity of central mineralocorticoid receptors. The treatment with GG extract or its active components may, therefore, be a useful adjunct therapy for patients with subtypes of depression and anxiety disorders with heightened renin–angiotensin–aldosterone system and/or inflammatory activity.

## 1. Introduction

Major depression is a heterogeneous disorder, which responds differentially to antidepressant treatment. Increased plasma concentrations of aldosterone have been observed in patients with depression [1,2,3,4]. Moreover, we identified a pattern of biological changes, which is associated with lower responsiveness to antidepressive therapy, which include a high aldosterone/cortisol ratio, a lower systolic blood pressure, and higher salt preference [5,6]. These markers indicate a disruption in aldosterone–mineralocorticoid (MR) signaling. Modulation of aldosterone signaling forms the theoretical framework for the present study design.

Higher salt intake and preference may be the consequence of higher central MR activation by aldosterone, its primary ligand. Aldosterone is a main driver (together with angiotensin II) of a higher salt intake, which is associated with a higher salt taste threshold and higher salt liking/appetite [7,8]. A higher salt taste threshold is associated with higher trait anxiety levels [9] and subjects who show higher levels of anxiety and depression symptoms consume higher amounts of salt [10,11]. Furthermore, monoamine depletion in healthy subjects was found to induce a parallel increase in salt taste threshold and anxiety [12]. Therefore, aldosterone may be a common mediator of anxiety and salt taste desensitization, which goes along with a higher salt intake.

Next to alterations in renin–angiotensin–aldosterone system (RAAS) activity, brain morphological changes, i.e., an increased choroid plexus and reduced corpus callosum volume were observed in patients with low responsiveness to antidepressive therapy, which points to changes in barrier function in some forms of depression. These morphological changes are related to inflammatory and metabolic processes [13]. Interestingly, major depression was associated with a reduced blood–brain barrier (BBB) permeability for some compounds [14]. However, other studies more commonly reported an increase in BBB permeability with inflammatory challenges [15,16,17]. Overall, insufficient information is available on the association of mental or behavioral functions and proteins that are crucial for barrier integrity. The main regulators of selective barrier permeability are several claudins together with other barrier proteins [18]. There is in fact evidence on barrier protein involvement in the pathophysiology of several psychiatric disorders [19]. However, very little information is available on the association of mental or behavioral functions and proteins that are crucial for barrier integrity.

One of the possibilities to influence aldosterone signaling is to inhibit the enzyme 11β -hydroxysteroid-dehydrogenase type 2 (11β-HSD2), a key enzyme necessary for the specificity of MR to aldosterone [20]. One of the 11β-HSD2 inhibitors is glycyrrhizin, an active component of licorice (*Glycyrrhiza glabra*). Inhibition of 11β-HSD2 results in a reduction in RAAS activity and an increase in blood pressure in animal models [21,22] and humans [21,23,24]. Treatment with glycyrrhizin induced antidepressant effects in two small studies in depressed patients [25,26]. Glycyrrhizin also has direct anti-inflammatory properties by blocking toll-like receptor 4 (TLR4) [27,28] and consequently protects neuronal function, potentially by influencing BBB [29,30,31] and white matter integrity [32] via claudins and other markers of the BBB.

The primary hypothesis of the present study is that food enrichment with an extract of *Glycyrrhiza glabra* in an animal model of chronic stress results in decreased anxiety behavior and reduced salt preference. The secondary hypothesis is that the mentioned changes are associated with altered gene expression of barrier proteins in the prefrontal cortex, which are downstream of inflammation mediators and associated with BBB function, and white matter integrity [33,34]. Additional parameters obtained in several tissues from the rats involved in the present study related to the cellular entry point of SARS-CoV-2 have been analyzed and published separately [35] because of the acute needs of the COVID-19 pandemic and their potential relevance in that context.

## 2. Material and Methods

### 2.1. Animals

The Animal Health and Animal Welfare Division of the State Veterinary and Food Administration of the Slovak Republic approved all experimental procedures (permission No. Ro 2291/18-221/3), which were conducted in accordance with the NIH Guidelines for Care and Use of Laboratory Animals. The experiments were performed on 48 Sprague-Dawley rats (Velaz, Prague, Czech Republic), baseline weight 225–250 g. After 5 days of acclimatization to standard laboratory conditions (two animal per cage, unlimited access to food and water, a 12:12 h light–dark cycle (light on from 07.00 h to 19.00 h), temperature of 22 ± 2 °C and humidity at 55 ± 10%), the rats were randomly assigned to the control groups (n = 24) and to groups of animals exposed to a chronic mild stress paradigm (n = 24).

Chronic mild stress was induced by different stress stimuli (Table 1). The stimuli lasted 12 h each, in a randomized order, i.e., two conditions per day for 5 weeks as described previously [35]. All stimuli were applied in the housing facility. Control animals were housed undisturbed in a different room under the same light and temperature conditions.

### 2.2. Treatment

Control and stress-exposed rats were randomly assigned to one of the two groups. One group was fed a diet with an extract of *Glycyrrhiza glabra* (n = 12); the other group was fed a standard diet (n = 12). The extract of *Glycyrrhiza glabra* roots (Gall-Pharma GmbH, Judenburg, Austria) (Batch. no. P17092209) contained 6.25% of glycyrrhizinic acid (glycyrrhizin). Water was used as a solvent during the extraction. The extract was mixed into the standard diet to achieve a dose of 150 mg/kg/day (SSNIFF Spezialdiäten GmbH, Soest, Germany). The dose was selected according to results of others [36], as described previously [35]. The standard diet (SSNIFF Spezialdiäten GmbH, Soest, Germany) consisted of carbohydrates (65%), protein (24%), and fat (11%). All animals received normal control diet for the first 3 weeks (Figure 1). The rats assigned for *Glycyrrhiza glabra* were fed the experimental diet for week 4–5 [35].

### 2.3. Behavioral Testing

#### 2.3.1. Elevated Plus-Maze Test

On day 31 of chronic mild stress, all animals underwent the elevated plus-maze test for a duration of 5 min to evaluate changes in anxiety behavior [37]. The test was conducted during the light phase of the cycle. The animals were allowed to acclimatize to the conditions in the room where the test takes place for 30 min and were then placed in the center of the maze. During the tests, the behavior of the animals was recorded by a video camera and analyzed later by EthoVision XT (Noldus EthoVision, XT 9.0, Noldus Information Technology, Wageningen, The Netherlands). The number of entries, time spent in the open arms, and the ratio of open to total arm entries (open/total x100) were used as measures of the anxiety level.

#### 2.3.2. Salt Preference Test

Possible changes in salt sensitivity were assessed using the salt preference test according to the previous reports by others [38,39]. The test was performed from day 31 to day 33 of chronic mild stress. During the test, rats were given a free choice between two bottles, one with 0.3 M NaCl solution and the other one with tap water for 48 h. The animals were not deprived of water and food before and during the test. To avoid possible effects of side preference on drinking behavior, the position of the bottles was switched after 24 h. The consumption of salt solution was measured by weighing the bottles every 24 h. The absolute salt intake (g) and the relative salt intake (g/g body weight x100) were calculated.

#### 2.3.3. Forced Swim Test (FST)

On days 33 and 34 of chronic mild stress, the animals were tested in the FST. The procedure was originally described by Porsolt et al. [40] and modified by Hlavacova et. al. [41]. The rats were placed in glass tanks filled with water at 23 °C. Two swimming sessions were conducted. The first pre-test session lasted 15 min. Twenty-four hours later, rats were subjected to the 5-min test session (day 34). Both swimming sessions were conducted during the light phase of the cycle. Behavior was scored from the video-recordings for the whole test session. The percentage of time that the animal spent struggling, swimming, and floating (immobile) was evaluated.

#### 2.3.4. Novel Object Recognition Test

The novel object recognition test was performed according to Zamberletti et al. [42] on day 30 of chronic mild stress. The test was conducted during the light phase of the cycle. The animals were allowed to acclimatize to the conditions in the room where the test took place for 1 h. During the training session, each rat was placed into the corner of the arena and two identical objects were presented to the animal. The rat was allowed to freely explore the objects for 5 min. Upon completion, the rat was returned to its home cage. Three min later the rat was returned to the arena and the test session was conducted. One of the familiar objects was replaced by a novel object. The rat was allowed to explore for 5 min. The movement of the rat was continuously video-recorded and the time spent in the exploration of the novel object and the familiar object was evaluated.

### 2.4. Organ Collection

At the end of week 5 of experimental procedures, the animals were quickly decapitated with a guillotine between 08.00 h and 10.30 h in the morning. The brain was quickly removed from the skull and the prefrontal cortex was dissected on an ice-cold plate. Subsequently, all samples were frozen and stored at −70 °C until they were analyzed.

### 2.5. Reverse Transcription and Quantitative Real-Time Polymerase Chain Reaction (RT-PCR)

The gene expression of the barrier proteins Zonula occludens protein-1 (ZO-1), Claudin-5, and Claudin-11 in the prefrontal cortex was measured by quantitative PCR. Total RNA was extracted using TRI Reagent^®^ according to the manufacturer’s protocol. Oligo(dT) primers were used to transcribe mRNA into cDNA with the use of M-MuLV reverse transcription system (ProtoScript, First Strand cDNA Synthesis Kit New England Biolabs, Ipswich, MA, USA). Primers specific for the studied genes as well as reference genes were designed by Primer BLAST NCBI software (Table 2). The geometric mean of HPRT1 and TfR1 reference genes was used to determine the relative gene expression of the studied genes. Gene expression was quantified with the use of QuantStudio5 system (Applied Biosystems^®^, Waltham, MA, USA) with the use of Luna^®^ qPCR Master Mix (New England Biolabs, Ipswich, MA, USA) as described previously [43,44].

### 2.6. Glial Fibrillary Acidic Protein (GFAP) Quantification by ELISA

The tissue of the frontal cortex was weighed and homogenized in phosphate-buffered saline (tissue weight (g): solution (mL) volume = 1:9 as recommended by the manufacturer’s protocol). Subsequently, the samples were sonicated and frozen during the night. Homogenates were centrifuged for 5 min at 5000× *g* to obtain the supernatant. All samples were diluted 50-fold to a working concentration with phosphate-buffered saline. Concentrations of GFAP protein were measured by an enzyme-linked immunosorbent assay (ELISA) kit (Antibodies-Online Inc., Pottstown, PA, USA). The detection limit of the assay was 62.5 pg/mL. The intra- and inter-assay coefficients of variation were 10% and 12%, respectively.

### 2.7. Statistical Analysis

The software package used for the statistical analysis was Statistica 7 (Statsoft, Tulsa, OK, USA). The authors were blinded to the experimental protocol while performing the experiments. The values were checked for normality of distribution using the Shapiro–Wilks test. Data not normally distributed, namely data on the % of time investigating the new object in the novel object recognition test, were ln transformed and then successfully checked for distributional properties by Shapiro–Wilk’s test. Data for claudin-5 gene expression were winsorized using a 15% two-tailed quantile trimming to treat the identified extreme outlying observations (1.5 × interquartile range rule). Winsorizing was needed in two data points, one in the control-*Glycyrrhiza glabra* group, one in the stress-*Glycyrrhiza glabra* group. All data were analyzed by two-way analysis of variance (ANOVA) with main factors of treatment (*Glycyrrhiza glabra* extract vs. placebo) and stress (chronic mild stress vs. control), as all the conditions for the use of this appropriate statistical method were fulfilled. For post hoc comparisons, the Tukey post hoc test was chosen as this test is stricter in comparison with other tests, such as Fisher least significant difference (LSD). Results are expressed as means ± standard error of the mean (SEM). The overall level of statistical significance was set as p < 0.05. The figures were created in GraphPad Prism 8 software (Dotmatics, Boston, MA, USA).

## 3. Results

### 3.1. Stressfulness of the Model Used

The exposure to chronic mild stress resulted in reduced body weight gain (Figure 2a). Two-way ANOVA revealed a significant main effect of stress (F(_1,42_) = 45.20, p < 0.001) on body weight gain. No significant main effect of treatment or interaction was found.

The protein concentrations of GFAP in the frontal cortex were not affected by *Glycyrrhiza glabra* extract treatment. There was a numerical reduction in GFAP protein concentrations in rats exposed to chronic mild stress compared to those in unstressed rats, which failed to be statistically significant (F(_1,44_) = 0.13, p = 0.053). The effect of interaction was not statistically significant.

Exposure to chronic mild stress reduced time spent in struggling behavior (F(_1,44_) = 5.14, p < 0.05) and increased swimming behavior (F_(1,44)_ = 5.01, p < 0.05) in stressed animals compared to unstressed rats (Figure 2b,c). Immobile behavior was not affected by chronic stress (Figure 2d). The effect of treatment with extract of *Glycyrrhiza glabra* or interaction was not statistically significant.

### 3.2. Anxiety Behavior

The diet enriched with *Glycyrrhiza glabra* extract resulted in reduced anxiety behavior measured in the elevated plus-maze test. Statistical analysis of spatiotemporal measures revealed a significant main effect of treatment on the number of open arm entries (F(_1,44_) = 13.66, p < 0.001) and the time spent in the open arms (F(_1,44_) = 11.49, p < 0.01). Animals fed the diet with *Glycyrrhiza glabra* extract entered significantly more often (Figure 3a) and spent more time in the open arms of the maze (Figure 3b). As revealed by a two-way ANOVA, a significant main effect of treatment was observed also in locomotor activity. Rats fed the diet with *Glycyrrhiza glabra* extract exhibited a significantly increased locomotion activity compared to rats fed the placebo diet (F(_1,44_) = 4.39, p < 0.05) (Figure 3c). The ratio of open to total arm entries was significantly affected by the treatment. *Glycyrrhiza-glabra*-treated animals showed a significantly increased ratio of open to total arm entries compared to those fed the placebo diet (F(_1,44_) = 8.54, p < 0.01) (Figure 3d). There was no significant main effect of stress or significant interaction on open arms entries and time spent in the open arms or locomotion activity as well as on the ratio of open to total arm entries.

### 3.3. Cognitive Testing

The diet enriched with *Glycyrrhiza glabra* extract and exposure to chronic mild stress did not affect behaviors in the novel object recognition test. Two-way ANOVA did not show a significant difference in the time spent in the exploration of the novel object and the familiar object between the groups (Figure 4).

### 3.4. Salt Preference

With respect to salt preference and relative salt intake on day 1 of the test, two-way ANOVA showed significant interactions between the main factors of treatment and stress exposure in salt preference (F(_1,44_) = 8.09, p < 0.01) as well as relative salt intake (F(_1,43_) = 7.85, p < 0.01). The post hoc analysis showed a significant decrease in unstressed rats, which received *Glycyrrhiza glabra* extract vs. placebo diet in salt preference (p < 0.05; Figure 5a) and relative salt intake (p < 0.05; Figure 5c). No significant main effect of stress or interaction with treatment was detected.

With respect to data obtained on day 2 of the salt preference test (Figure 5b), two-way ANOVA showed significant main effects of treatment (F(_1,44_) = 11.08, p < 0.01), as well as significant treatment x stress interaction (F(_1,44_) = 6.85, p < 0.05). Post hoc analysis revealed that salt preference was significantly decreased in the unstressed rats fed the diet with *Glycyrrhiza glabra* extract compared to those fed the placebo diet (p < 0.001). No significant main effect of stress was observed. Statistical analysis by a two-way ANOVA revealed a significant main effect of treatment (F(_1,44_) = 10.55, p < 0.01) as well as treatment x stress interaction (F(_1,44)_ = 6.66, p < 0.05) on relative salt intake on the second day of the test (Figure 5d). In unstressed rats, Tukey post hoc analysis revealed significantly lower relative salt intake in rats treated with *Glycyrrhiza glabra* extract compared to those with placebo diet (p < 0.001). No significant main effect of stress was found.

### 3.5. Gene Expression of Barrier Proteins

The gene expression of ZO-1 in the prefrontal cortex was affected by treatment (Figure 6a). Two-way ANOVA revealed that the concentrations of mRNA coding for ZO-1 was significantly lower in animals treated with *Glycyrrhiza glabra* extract when compared to controls (F_(1,42)_ = 5.6972; p < 0.05).

No main effect of stress or treatment was observed for the concentrations of claudin-11 mRNA in the prefrontal cortex (Figure 6b). However, two-way ANOVA revealed a significant interaction between stress and *Glycyrrhiza glabra* treatment (F_(1,42)_ = 12.899; p < 0.001). Tukey post hoc analysis shows that unstressed but not stressed animals receiving the *Glycyrrhiza glabra* treatment had higher concentrations of claudin-11 when compared with the control group. Stress by itself led to an increase in claudin-11 gene expression in the untreated group, which was prevented by *Glycyrrhiza glabra.* The exposure to chronic mild stress did not affect the gene expression of claudin-5 in the prefrontal cortex. Similarly, no effect of the diet enriched with *Glycyrrhiza glabra* extract was observed.

## 4. Discussion

This study was designed to simulate an intervention with *Glycyrrhiza glabra* extract in individuals that were already suffering from chronic stress, not as a preventive measure against chronic stress. The results show an anxiolytic effect of chronic food enrichment with an extract of *Glycyrrhiza glabra* in both stressed and non-stressed animals. Salt preference and salt intake were significantly reduced by treatment with *Glycyrrhiza glabra* under control, but not stress conditions. The gene expression of the barrier protein claudin-11 in the prefrontal cortex was increased in unstressed rats exposed to *Glycyrrhiza-glabra*-enriched diet. The stress-induced rise in cortical claudin-11 expression was absent in stressed *Glycyrrhiza glabra* treated animals. Exposure to *Glycyrrhiza glabra* enriched diet resulted in a decrease in concentrations of mRNA coding for ZO-1 irrespective of the stress and non-stress conditions.

How these alterations are mediated has not been completely resolved. Only a few areas in the brain are sensitive to aldosterone, in particular the nucleus of the solitary tract (NTS), which co-expresses MR and 11β-HSD2 [45,46]. This enzyme provides specificity to aldosterone by quickly degrading cortisol/corticosterone, which competes with aldosterone [20,47]. The NTS structure is crucial for autonomic and affect regulation [48,49], including salt appetite [7,8,50,51,52]. Projections of the NTS to higher brain areas, in particular the amygdala and prefrontal cortex, appear to be involved in the association between NTS activity and affective states [53,54].

Several parameters were used to assess the stressfulness of the procedure. The stressfulness of the chronic mild stress model used here was confirmed by reduced body weight gain and food intake. Consistently, concentrations of GFAP, a protein responsible for the maintenance of glial cells and supporting the BBB [55] were marginally decreased (p = 0.053) in the frontal cortex of stressed animals.

The time spent in immobility, often considered a sign of depression-like behavior, was not increased in rats exposed to the chronic mild stress model, though the struggling behavior was significantly reduced. It has been recently discussed that the time spent in immobility is aligned to cognitive functions underlying behavioral adaptation rather than depression-like behavior [56,57].

In this context, an antidepressive effect of *Glycyrrhiza glabra* should not be ruled out. Indeed, depression is often comorbid with increased anxiety and is then referred to as anxious depression. There is evidence that anxious depression is more refractory to standard treatment than non-anxious depression [58]. In the present study, the treatment with *Glycyrrhiza glabra* extract substantially reduced anxiety behavior in the elevated plus-maze test irrespective of the stressor exposure and may indicate specificity for this depression subtype. Anxiolytic and antidepressant effects of *Glycyrrhiza glabra* extract were already observed in animal [59,60,61,62] and human [25,26] studies, encouraging future drug development.

The present study revealed that chronic treatment with *Glycyrrhiza glabra* extract induced a reduction of salt intake and salt preference under non-stress conditions. Such an effect of the extract may be attributed to a reduction in RAAS activity, plasma aldosterone levels, and, as a consequence, an associated decrease in central MR activity. This may lead to the observed decrease in anxiety behavior. An alternative hypothesis that the observed lower salt intake reduces anxiety is conceivable but not very likely. Namely, salt depletion in animal models [48,63,64,65,66] and humans [67,68] is associated with higher levels of anxiety and depression, which is in line with the fact that low salt diet leads to an increase in RAAS activity [69].

However, we did not observe a direct change in aldosterone concentrations by the extract as measured in the same animals, but a reduced aldosterone/corticosterone ratio, expressed as a significantly increased corticosterone concentration [35]. The analogue ratio in humans, i.e., aldosterone/cortisol saliva ratio, was predictive for therapy refractoriness in patients with depression [5]. As both the release of cortisol/corticosterone and aldosterone is stimulated by Adrenocorticotropic hormone, an increase in this ratio signifies an increase in the activity of renin–angiotensin–aldosterone cascade, as the effect of adrenocorticotropic hormone is corrected for. It may be speculated that angiotensin II, the trigger hormone to release aldosterone, has a direct effect, which is synergistic to that of aldosterone, for example at the level of the NTS [45,70], and has anxiogenic activity by itself [71,72].

The measurements of the expression of barrier proteins in the present study was motivated by previously published indices on BBB changes under chronic stress in animals [17] and in patients with major depression [73]. The latter authors demonstrated a strong association between increased peripheral inflammation indexed by *C*-reactive protein and altered BBB permeability to the imaging ligand PK11195 [73]. We and others have outlined that central nervous system inflammation may be mediated via the choroid plexus, which may release inflammatory mediators [74,75]. Interestingly, higher inflammatory activity and choroid plexus volume in patients with major depression are associated with an altered BBB permeability [14]. Indeed, increased inflammatory activity has also been linked to an increased choroid plexus volume in neurological and psychiatric conditions [14,76,77,78,79].

The exposure to chronic mild stress resulted in a rise in concentrations of mRNA coding for claudin-11 in the brain cortex. To our knowledge, there is no information on any changes in brain claudin-11 expression under stress conditions in the literature available. An important finding of the present study is the increased gene expression of claudin-11 in the prefrontal cortex of unstressed rats exposed to a *Glycyrrhiza-glabra*-enriched diet. The present study further shows that the stress-induced rise in cortical claudin-11 expression was absent in stressed *Glycyrrhiza-glabra*-treated animals.

Claudin-11 is a very crucial component of the neuronal myelin sheaths in white matter and affects neurotransmission [80]. There are observations of an improved myelination induced by glycyrrhizin in an animal model of multiple sclerosis, which appears to be based on its anti-inflammatory properties [32]. This, however, implies that the studied sections of prefrontal cortex contain white matter tissue. In addition, the suppression of the renin–angiotensin–aldosterone system and, therefore, assumed reduction in angiotensin II and aldosterone may also have direct white matter protective properties, as demonstrated in a hypoxia model in rats [81]. This needs to be further studied. Regarding peripheral tissues, there is a study investigating the blood–testis barrier, which revealed a decrease in claudin-11 expression in the Sertoli cells under chronic stress [82].

The exposure to chronic mild stress failed to modify the gene expression of barrier protein ZO-1. In another stress model, chronic isolation in our previous study led to an increase in ZO-1 and a decrease in claudin-5 in the prefrontal cortex [43], i.e., an opposite response of tight junction proteins in response to stress. The mentioned study described similar findings in the intestine. Sun et al. [83] found that in mice in a chronic mild stress model, depression-like states were accompanied by hippocampal BBB breakdown and claudin-5 downregulation. Antidepressant treatment reversed these changes. The present results on claudin-5 gene expression show that this is not the case in the brain prefrontal cortex.

The investigation of a *Glycyrrhiza-glabra*-enriched diet on markers of BBB integrity revealed a decrease in gene expression of ZO-1 irrespective of the stress and non-stress conditions. Such a decrease was, however, not observed in isolated porcine BBB endothelial cells while investigating the effects of a glycyrrhizin metabolite 18β-glycyrrhetinic acid [84]. This points to an indirect mediation of *Glycyrrhiza glabra* extract affecting the barrier integrity. The mechanistic impact of this change is, however, not clear yet. A clue to these changes may be that ZO-1 appears to be under the influence of angiotensin II, the aldosterone-releasing compound as part of the RAAS, which suppresses the expression of ZO-1 in brain endothelial cells [85,86], and which may have an effect in the current model to suppress ZO-1 expression as well.

A proinflammatory effect of aldosterone has been described in several tissues [41,87,88,89,90,91,92,93], including capillary endothelium [94]. Aldosterone acts synergistically at the innate immunity receptor, TLR4, with the TLR4 ligand lipopolysaccharide (LPS) to induce depression-like behavior [95]; therefore, the combined action of *Glycyrrhiza glabra* to reduce the RAAS [21,22,23,24] and antagonize the TLR4 receptor acts synergistically to reduce inflammation.

## 5. Conclusions

In conclusion and in line with our hypotheses, the treatment with *Glycyrrhiza glabra* extract had beneficial behavioral effects, particularly on anxiety. These effects were paralleled by a reduction in salt appetite along with modulation of the gene expression of two barrier proteins participating in the regulation of BBB and white matter integrity. The treatment with *Glycyrrhiza glabra* extract or its active components may, therefore, be a useful adjunct therapy for patients with subtypes of depression and anxiety disorders with heightened RAAS and/or inflammatory activity.

## Figures and Tables

**Figure 1 nutrients-16-00515-f001:**
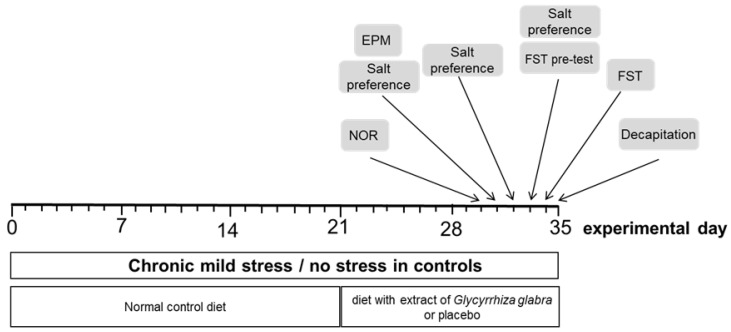
The time course of the study. EPM = elevated plus maze; NOR = novel object recognition; FST = forced swim test.

**Figure 2 nutrients-16-00515-f002:**
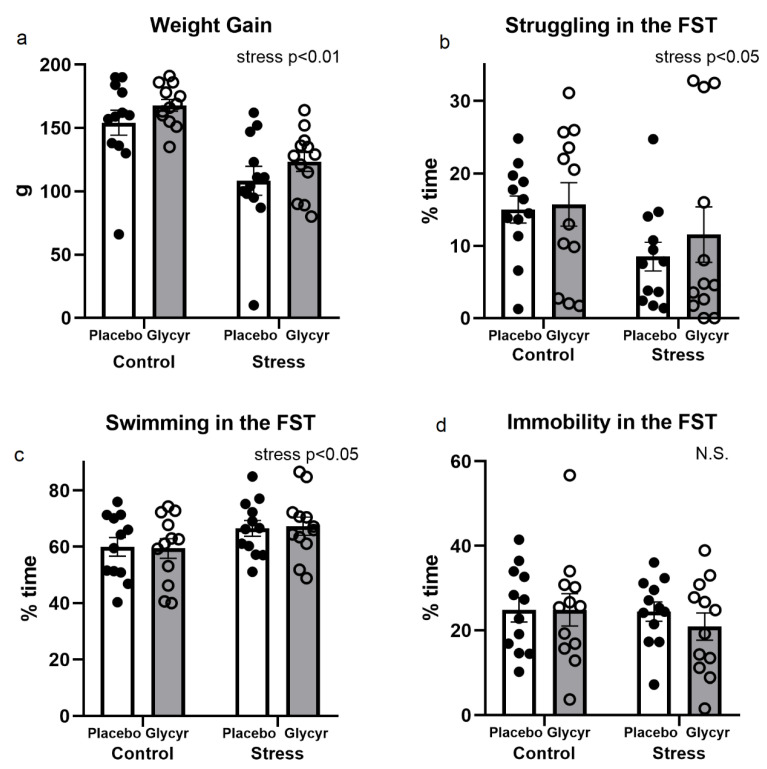
Parameters showing the stressfulness of the model used: animal weight gain (**a**); struggling behavior in the forced swim test (FST) (**b**); swimming behavior in the FST (**c**); immobility in the FST (**d**). N.S.: not significant.

**Figure 3 nutrients-16-00515-f003:**
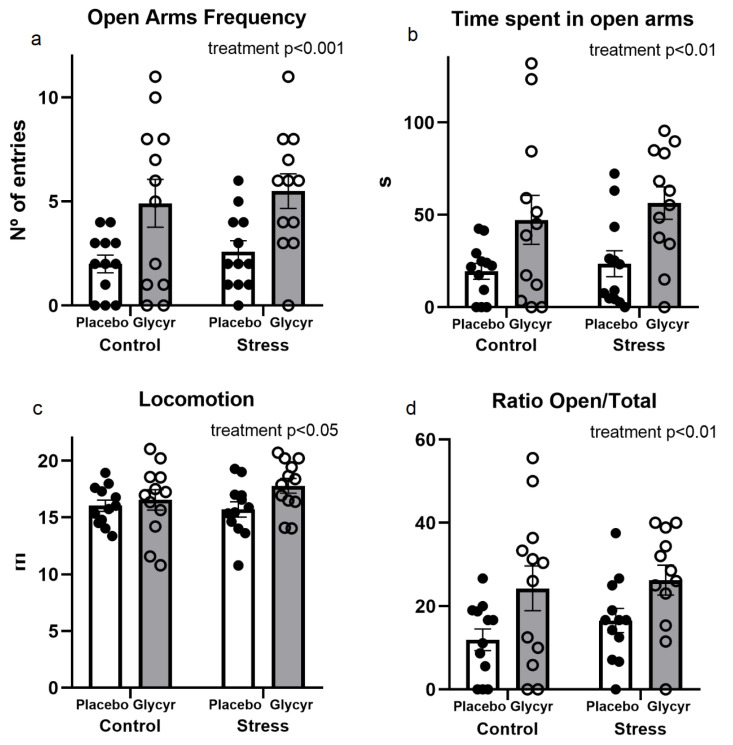
The effects of chronic mild stress and *Glycyrrhiza glabra* extract treatment on anxiety behaviors measured in the elevated plus-maze test: frequency of entering into the open arms of the maze (**a**); time spent in the open arms (**b**); locomotion (**c**); and ratio of entries into the open arms and total entries (**d**). Each value represents the mean ± SEM (n = 12 rats per group). Statistical significance as revealed by two-way ANOVA.

**Figure 4 nutrients-16-00515-f004:**
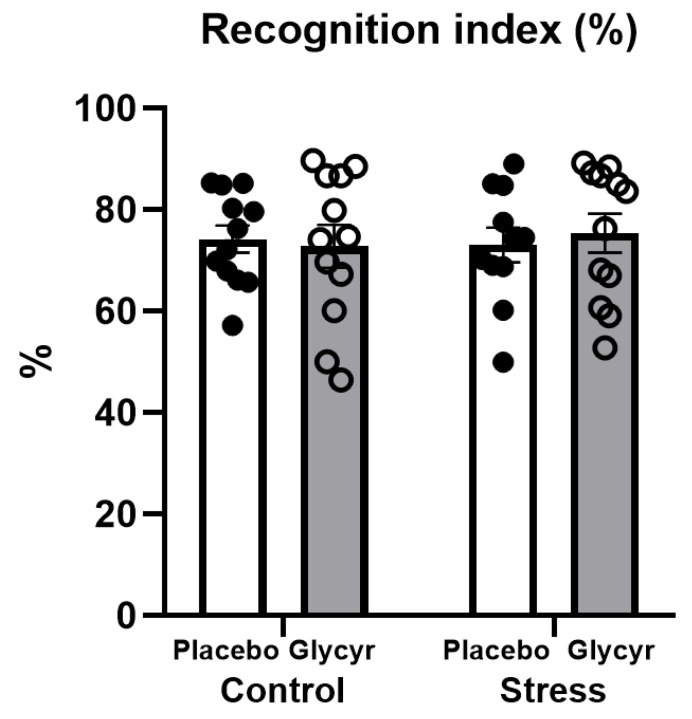
The effects of chronic mild stress and *Glycyrrhiza glabra* extract treatment on behavior in the novel object recognition test. Each value represents the mean ± SEM (n = 12 rats per group). Statistical significance as revealed by two-way ANOVA.

**Figure 5 nutrients-16-00515-f005:**
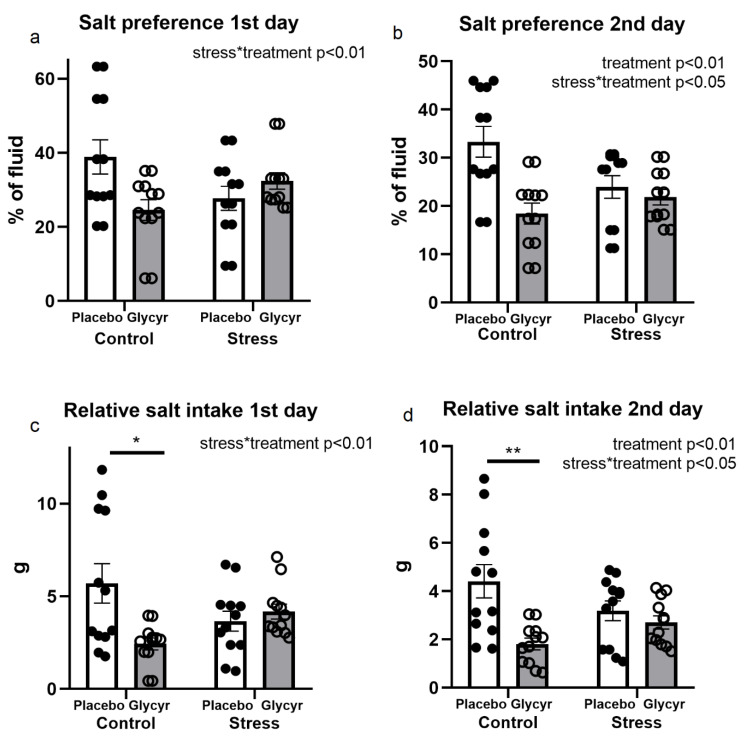
Salt preference on days 1 (**a**) and 2 (**b**) of the salt preference test with respective relative salt intakes (**c**,**d**). Each value represents the mean ± SEM (n = 12 rats per group). Statistical significance as revealed by two-way ANOVA. (*: p < 0.05; **: p < 0.01)

**Figure 6 nutrients-16-00515-f006:**
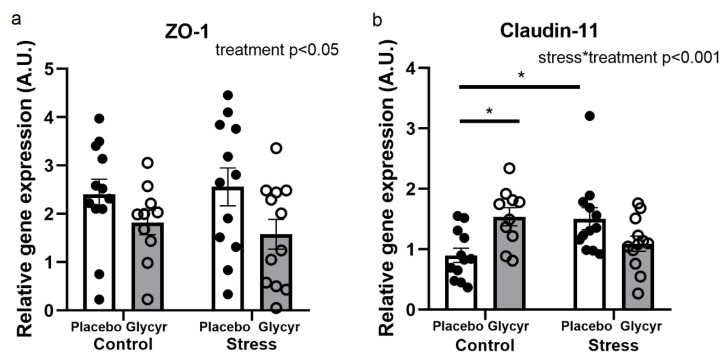
Relative gene expression of ZO-1 (**a**) and claudin-11 (**b**) in the prefrontal cortex. Each value represents mean ± standard error of the mean (SEM) (n = 12 rats/group). Statistical significance as revealed by two-way ANOVA. (*: p < 0.05)

**Table 1 nutrients-16-00515-t001:** List of stress situations.

Stressor	Stressor Description
*Social isolation*	Alone in the cage
*Unknown cage-mate*	Sharing the cage with an unknown rat
*Stroboscopic light*	Light flashes with a frequency of 5 flashes/s
*Cage tilt*	Cages tilted by 45 degrees
*Wet cage*	Water surface 2 cm above the bottom of the cage
*Continuous lighting*	Lights on for 24 h
*Water deprivation*	Without water for 12 h
*White noise*	Sound of 90 dB for 12 h
*Uncomfortable cage*	Cage without bedding

**Table 2 nutrients-16-00515-t002:** Primer sequences used in qPCR analyses.

Gene	Sense	Sequence 5′ → 3′
ZO-1	Forward	CATGAGAAGCAGACACCCACT
	Reverse	CAGTTTCATGCTGGGCCTAA
Claudin-5	Forward	CGCTTGTGGCACTCTTTGT
	Reverse	ACTCCCGGACTACGATGTTG
Claudin-11	Forward	ATTGGCATCATCGTCACAAC
	Reverse	ATGTCCACCAGGGGCTTG
HPRT1	Forward	CGTCGTGATTAGTGATGATGAAC
	Reverse	CAAGTCTTTCAGTCCTGTCCATAA
TfR1	Forward	ATACGTTCCCCGTTGTTGAGG
	Reverse	GGCGGAAACTGAGTATGGTTGA

## Data Availability

Data are available upon request.

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
