# Peer review of "Treatment with Glycyrrhiza glabra Extract Induces Anxiolytic Effects Associated with Reduced Salt Preference and Changes in Barrier Protein Gene Expression"

_nutrients, 2024, doi:10.3390/nu16040515_

Round 1

Reviewer 1 Report

Comments and Suggestions for Authors

The paper by Murck and collaborators investigates the antidepressant effects of a Glycyrrhiza glabra (GG) extract in male Sprague-Dawley rats exposed to chronic mild stress for five weeks. Authors showed that GG extract induced anxiolytic effects, reduced salt preference and salt intake, and increased gene expression of the barrier protein claudin-11 in the prefrontal cortex. Moreover, exposure to GG resulted in ZO-1-expression, suggesting the role of GG extract in the management of depression or anxiety disorders. This paper deals with an interesting subject and has an appropriate level of novelty, yet several flaws are present. Therefore, the paper should be thoroughly revised before being considered for publication in Nutrients.

1)     The whole formatting of the paper must be revised according to MDPI’s guidelines (i.e., headings, citations, etc). In addition, the punctuation and phrasing must be checked throughout the text.

2)     In my opinion, the data presented in the supplementary files should be reported directly in the text since, especially for the stress given to rats, it is quite relevant to see directly when reading the text. Supplementary are thought to be elements that are marginally important for the comprehension of the work.

3)     The Abstract should present just the relevant information. The first lines are not so important to understand the context (see also the following comment). Therefore, Authors are suggested to improve it.

4)     The Introduction is meant to give a broad view of the setting of the study, to lead the readers understand the basis of the aims. In the case of this paper, this section resembles what is meant to be done for the Discussion. Therefore, Authors should rewrite the Introduction to be simpler and to give a clear and direct context to support the aim.

5)     Why do the Authors choose to give the GG extract only at the two final weeks?

6)     The description of GFAP expression should be dealt separately. There is no apparent reason to be discussed with FST results. This is the same for weight gain.

7)     Cognition results must be also shown, despite the absence of significance.

8)     Discussion should me improved because it is not sufficient to describe the results with a hint of the related literature. Here, Authors should bring the readers to understand the true meaning of their work and its significance.

Comments on the Quality of English Language

The English language is used appropriately, yet punctuation needs a thorough revision. The use of linking words is advised to render the reading more fluent.

Author Response

1)     The whole formatting of the paper must be revised according to MDPI’s guidelines (i.e., headings, citations, etc). In addition, the punctuation and phrasing must be checked throughout the text.

Response

The formatting of the paper has been corrected.

2)     In my opinion, the data presented in the supplementary files should be reported directly in the text since, especially for the stress given to rats, it is quite relevant to see directly when reading the text. Supplementary are thought to be elements that are marginally important for the comprehension of the work.

Response

The former supplementary data are now reported in the text.

3)     The Abstract should present just the relevant information. The first lines are not so important to understand the context (see also the following comment). Therefore, Authors are suggested to improve it.

Response

4)     The Introduction is meant to give a broad view of the setting of the study, to lead the readers understand the basis of the aims. In the case of this paper, this section resembles what is meant to be done for the Discussion. Therefore, Authors should rewrite the Introduction to be simpler and to give a clear and direct context to support the aim.

Response

The Introduction has been rewritten in the revised version of the manuscript. It is now shorter, simpler, and is in a more direct context to support the aims.

5)     Why do the Authors choose to give the GG extract only at the two final weeks?

Response

We are extremely thankful for this important comment.

The following statement has been added as the first sentence of the Discussion:

The study was designed to simulate an intervention with Glycyrrhiza glabra extract in individuals that were already suffering from chronic stress, not as a preventive measure against chronic stress.

6)     The description of GFAP expression should be dealt separately. There is no apparent reason to be discussed with FST results. This is the same for weight gain.

Response

Following reviewer suggestions, the text has been modified as follows:

Several parameters were used to assess the stressfulness of the procedure used. The stressfulness of the chronic mild stress model used was confirmed by reduced body weight gain and food intake. Consistently, concentrations of GFAP, a protein responsible for the maintenance of glial cells and supporting the blood-brain barrier (Yang and Wang 2015) were marginally decreased (p=0.053) in the frontal cortex of stressed animals.

The time spent in immobility, often considered a sign of depression-like behavior, was not increased in rats exposed to the chronic mild stress model, though the struggling behavior was significantly reduced. It has been recently discussed that the time spent in immobility is aligned to cognitive functions underlying behavioral adaptation rather than depression-like behavior (Molendijk and de Kloet, 2015, 2022).

7)     Cognition results must be also shown, despite the absence of significance.

Response

The diet enriched with Glycyrrhiza glabra extract and exposure to chronic mild stress did not affect behavior in the novel object recognition test. Two-way ANOVA did not show a significant difference in the time spent in the exploration of the novel object and the familiar object between the groups. The recognition index is shown on Fig. X.

8)     Discussion should me improved because it is not sufficient to describe the results with a hint of the related literature. Here, Authors should bring the readers to understand the true meaning of their work and its significance.

Response

We thank the reviewer for encouraging us to improve the Discussion. We did our best to discuss the results more deeply.

Comments on the Quality of English Language

The English language is used appropriately, yet punctuation needs a thorough revision. The use of linking words is advised to render the reading more fluent.

Response

The text has been corrected in agreement with the reviewer's suggestions.

Reviewer 2 Report

Comments and Suggestions for Authors

Positive aspects of the manuscript include a well-defined hypothesis linking depression, aldosterone, and Glycyrrhiza glabra extract.  The comprehensive literature review, clear study design, and rigorous behavioral and molecular analyses contribute to the study's scientific merit, holding potential for meaningful contributions to the field.

Suggestions for manuscripts:

1. Language and Clarity: The manuscript requires substantial language editing for clarity and fluency.   Some sentences are convoluted and may impede understanding. Consider rephrasing complex sentences to improve readability and coherence.

2. Redundancy in Results Presentation: There is redundancy in presenting results, particularly in the discussion of the impact of Glycyrrhiza glabra extract on salt preference and depression-related behaviors.   Streamlining this section could enhance focus.

3.  Statistical Analysis: While the statistical methods are mentioned, more details about specific statistical tests used and rationale for their selection would be beneficial. Consider providing a clear explanation of how the data were transformed or winsorized for non-normal distributions.

4. Connectivity between sections: Improve the flow between sections to ensure the readability from introduction to methods, results, and discussion. Please improve the logicality of the study to enhance overall coherence.

5. Depth of Discussion: The discussion would benefit greatly from a deeper exploration of the underlying mechanisms by which Glycyrrhiza glabra extract may exert its effects. Includes a more detailed explanation of how barrier protein changes related to the study's central hypothesis.

Overall, this study uses a sound approach to address a relevant and interesting research question. With improvements in the language, data presentation, and further elaboration of the discussion, this manuscript has the potential to make a significant scientific contribution.

Comments on the Quality of English Language

The language expression of the manuscript is basically accurate and the quality is good. It is recommended to simplify the presentation so that readers can better understand the meaning of the article.

Author Response

Positive aspects of the manuscript include a well-defined hypothesis linking depression, aldosterone, and Glycyrrhiza glabra extract.  The comprehensive literature review, clear study design, and rigorous behavioral and molecular analyses contribute to the study's scientific merit, holding potential for meaningful contributions to the field.

Response

We thank the reviewer for his/her positive feedback.

Suggestions for manuscripts:

  1. Language and Clarity: The manuscript requires substantial language editing for clarity and fluency. Some sentences are convoluted and may impede understanding. Consider rephrasing complex sentences to improve readability and coherence.

Response

We did our best to make corrections as suggested.

  1. Redundancy in Results Presentation: There is redundancy in presenting results, particularly in the discussion of the impact of Glycyrrhiza glabra extract on salt preference and depression-related behaviors. Streamlining this section could enhance focus.

Response

Several redundancies have been removed.

  1. Statistical Analysis: While the statistical methods are mentioned, more details about specific statistical tests used and rationale for their selection would be beneficial. Consider providing a clear explanation of how the data were transformed or winsorized for non-normal distributions.

Response

We agree with the reviewer. More details about statistical tests are provided in the revised version of the manuscript.

The following changes have been made in the Methods:

The software package used for the statistical analysis was Statistica 7 (Statsoft, Tulsa, OK, USA). The authors were blinded to the experimental protocol while performing the experiments. The values were checked for normality of distribution using the Shapiro-Wilks test. Data not normally distributed, namely data on the % of time investigating the new object in the novel object recognition test, were ln transformed and then successfully checked for distributional properties by Shapiro Wilk's test. Data for claudin-5 gene expression were winsorized using a 15% two-tailed quantile trimming to treat the identified extreme outlying observations (1.5 × interquartile range rule). Winsorizing was needed in two data points, one in the Control-GG group, one in the Stress-GG group. All data were analyzed by two-way analysis of variance (ANOVA) with main factors of treatment (Glycyrrhiza glabra extract vs. placebo) and stress (chronic mild stress vs. control), as all the conditions for the use of this appropriate statistical method were fulfilled. For post hoc comparisons, the Tukey post hoc test was chosen as this test is stricter in comparison with other tests, such as Fisher least significant difference (LSD). Results are expressed as means ± standard error of the mean (SEM). The overall level of statistical significance was set as p < 0.05. The figures were created in GraphPad Prism 8 software (Dotmatics, MA, USA).

  1. Connectivity between sections: Improve the flow between sections to ensure the readability from introduction to methods, results, and discussion. Please improve the logicality of the study to enhance overall coherence.

Response

We believe that the flow of the text in the revised manuscript has been improved.

  1. Depth of Discussion: The discussion would benefit greatly from a deeper exploration of the underlying mechanisms by which Glycyrrhiza glabra extract may exert its effects. Includes a more detailed explanation of how barrier protein changes related to the study's central hypothesis.

Response

We thank the reviewer for this important suggestion. Accordingly, a more detailed explanation of how barrier protein changes related to the study's central hypothesis has been added to the Discussion.

Overall, this study uses a sound approach to address a relevant and interesting research question. With improvements in the language, data presentation, and further elaboration of the discussion, this manuscript has the potential to make a significant scientific contribution.

Response

We appreciate this positive judgment.

Comments on the Quality of English Language

The language expression of the manuscript is basically accurate and the quality is good. It is recommended to simplify the presentation so that readers can better understand the meaning of the article.

Response

We did our best to correct the text accordingly.

Reviewer 3 Report

Comments and Suggestions for Authors

Treatment with Glycyrrhiza glabra extract induces anxiolytic effects associated with reduced salt preference and changes in barrier protein gene expression

 The study discusses the protective effect of Glycyrrhiza glabra extract and the anxiolytic effect and salt preference under salt stress which might have antidepressant effects. The inhibition of 11betaHSD2 by Glycyrrhiza glabra extract affects salt consumption and behavior via reducing aldosterone release.

The study is of significance in employing plant products/extracts for the treatment of anxiety but the manuscript needs to be revised substantially before being considered.

How the present study can be utilized for developing anxiolytic drugs? Discuss.

Is there any research on the development of antidepressant drugs from natural products? Please provide a note if documented.

Line 101-103: The primary hypothesis of the present study is that food enrichment with an extract of Glycyrrhiza glabra in an animal model of chronic stress results in decreased anxiety behavior and reduced salt preference.

However, as per the results obtained GG induced anxiolytic effects in both stressed and non-stressed animals, as measured in an elevated plus maze test. Moreover, Salt preference and salt intake were significantly reduced by Glycyrrhiza glabra-enriched diet under control, but not stress conditions.

There seems to be ambiguity between the hypothesis and the results of the study. Explain accordingly.

What is the role of the Claudin gene family in the brain cortex and correlate with antidepressant effect? Explain.

English language needs to be moderately revised.

Comments on the Quality of English Language

Moderate revisions for the English language, and grammatical errors are required.

Author Response

Treatment with Glycyrrhiza glabra extract induces anxiolytic effects associated with reduced salt preference and changes in barrier protein gene expression

 The study discusses the protective effect of Glycyrrhiza glabra extract and the anxiolytic effect and salt preference under salt stress which might have antidepressant effects. The inhibition of 11betaHSD2 by Glycyrrhiza glabra extract affects salt consumption and behavior via reducing aldosterone release.

The study is of significance in employing plant products/extracts for the treatment of anxiety but the manuscript needs to be revised substantially before being considered.

Response

We thank the reviewer for his/her positive feedback.

How the present study can be utilized for developing anxiolytic drugs? Discuss.

Is there any research on the development of antidepressant drugs from natural products? Please provide a note if documented.

Response

As suggested the observed anxiolytic has been further elaborated in the Discussion:

In this context, an antidepressive effect of Glycyrrhiza glabra should not be ruled out. Indeed, depression is often comorbid with increased anxiety and is then referred to as anxious depression. There is evidence that anxious depression is more refractory to standard treatment than non-anxious depression (Fava et al., 2008). In the present study, the treatment with Glycyrrhiza glabra extract substantially reduced anxiety behaviour in the elevated plus-maze test irrespective of the stressor exposure and may indicate specificity for this depression subtype. Anxiolytic, and antidepressant effects of Glycyrrhiza glabra extract were already observed in animal (Dhingra and Sharma, 2006; Hisaoka-Nakashima et al., 2019; Jiang et al., 2022; Wang et al., 2018) and human (Cao et al., 2020; Murck et al., 2020a) studies, encouraging future drug development.

Line 101-103: The primary hypothesis of the present study is that food enrichment with an extract of Glycyrrhiza glabra in an animal model of chronic stress results in decreased anxiety behavior and reduced salt preference.

However, as per the results obtained GG induced anxiolytic effects in both stressed and non-stressed animals, as measured in an elevated plus maze test. Moreover, Salt preference and salt intake were significantly reduced by Glycyrrhiza glabra-enriched diet under control, but not stress conditions.

There seems to be ambiguity between the hypothesis and the results of the study. Explain accordingly.

Response

This might be a small misunderstanding. In the hypothesis, we mention that we work with an animal model of stress and not that we expect changes only under stress conditions.

What is the role of the Claudin gene family in the brain cortex and correlate with antidepressant effect? Explain.

Response

We have added several pieces of information on this topic in the Discussion.

This is an example:

Sun et al.  [87] found that in mice in a chronic mild stress model, depression-like states were accompanied by hippocampal BBB breakdown and claudin-5 downregulation. Antidepressant treatment reversed these changes. The present results on claudin-5 gene expression show that this is not the case in the brain prefrontal cortex.    

English language needs to be moderately revised.

Response

We did our best to correct the text accordingly.

Round 2

Reviewer 1 Report

Comments and Suggestions for Authors

The Authors have answered to the elements pointed out. Therefore, the paper can be accepted for publication.

Comments on the Quality of English Language

The English language has been improved.

Reviewer 3 Report

Comments and Suggestions for Authors

Thank you for revising the manuscript.